# Epigenetics in Inflammatory Breast Cancer: Biological Features and Therapeutic Perspectives

**DOI:** 10.3390/cells9051164

**Published:** 2020-05-08

**Authors:** Flavia Lima Costa Faldoni, Cláudia Aparecida Rainho, Silvia Regina Rogatto

**Affiliations:** 1Department of Gynecology and Obstetrics, Medical School, São Paulo State University (UNESP), Botucatu 18618-687, São Paulo, Brazil; flaviafaldoni@gmail.com; 2Hermínio Ometto Foundation, Araras 13607-339, São Paulo, Brazil; 3Department of Chemical and Biological Sciences, Institute of Biosciences, São Paulo State University (UNESP), Botucatu 18618-689, São Paulo, Brazil; claudia.rainho@unesp.br; 4Department of Clinical Genetics, Lillebaelt University Hospital of Southern Denmark, Institute of Regional Health Research, University of Southern Denmark, 7100 Odense, Denmark

**Keywords:** inflammatory breast cancer, DNA methylation, histone modifications, histone deacetylases, epigenetic therapy

## Abstract

Evidence has emerged implicating epigenetic alterations in inflammatory breast cancer (IBC) origin and progression. IBC is a rare and rapidly progressing disease, considered the most aggressive type of breast cancer (BC). At clinical presentation, IBC is characterized by diffuse erythema, skin ridging, dermal lymphatic invasion, and peau d’orange aspect. The widespread distribution of the tumor as emboli throughout the breast and intra- and intertumor heterogeneity is associated with its poor prognosis. In this review, we highlighted studies documenting the essential roles of epigenetic mechanisms in remodeling chromatin and modulating gene expression during mammary gland differentiation and the development of IBC. Compiling evidence has emerged implicating epigenetic changes as a common denominator linking the main risk factors (socioeconomic status, environmental exposure to endocrine disruptors, racial disparities, and obesity) with IBC development. DNA methylation changes and their impact on the diagnosis, prognosis, and treatment of IBC are also described. Recent studies are focusing on the use of histone deacetylase inhibitors as promising epigenetic drugs for treating IBC. All efforts must be undertaken to unravel the epigenetic marks that drive this disease and how this knowledge could impact strategies to reduce the risk of IBC development and progression.

## 1. Introduction

Breast cancer (BC) is a complex and heterogeneous disease, characterized by distinct molecular profiles, histological types, and clinical features. Inflammatory breast cancer (IBC) is rare and the most aggressive type of BC. Its incidence varies in different regions around the world [1,2]. In the United States, IBC incidence showed a trend to increase before the turn of the 21st century [3], accounting for 1 to 5% of all BC diagnosed. Despite the low incidence, the mortality rate in Western countries accounts from 7 to 10%, which corresponds to approximately 4000 deaths per year in the United States [4]. The disease progresses rapidly and is associated with unfavorable outcomes. Although the introduction of multimodality therapy has improved the five-year overall survival rates [5], the prognosis is still poor.

Efforts have been made to establish molecular biomarkers and therapeutic targets to improve clinical management and, potentially, the overall survival of IBC patients. Nonetheless, its rarity, uncommon clinical presentation, and the overlapping of signs and symptoms with other diseases or conditions, like those associated with tissue inflammation or locally advanced BC (LABC), contribute to the delayed diagnosis and challenges in defining effective treatments. The clinical IBC features and the spreading of small tumor emboli make tumor sampling difficult, which restricts the biopsy size. The substantial discomfort and pain during the biopsy, mainly to get samples at multiple time points, further contributes to the limited number of molecular profiling studies in IBC samples [6]. The high metastatic rates and poor overall survival highlight the need to develop new approaches to improve the management of IBC patients [7].

Epigenetic mechanisms, such as DNA methylation, post-translational histone modifications, noncoding RNAs, and chromatin remodelers, are key regulators during developmental processes (including X chromosome inactivation and genomic imprinting) [8]. In normal cells, epigenetic modifications play an essential role in the chromatin structure remodeling and regulation of gene expression. In cancer cells, disruption of the epigenetic program is often associated with the aberrant expression of cancer driver genes [9,10,11]. Accumulating evidence suggests that epigenetic changes contribute to the origin and progression of breast cancer. Therefore, these patients may benefit from new epigenetic biomarkers for screening [12,13,14], prognosis [15,16], and therapeutic approaches complementary to the standard protocols used in the clinical practice [17,18]. In this review, we summarize epigenetic findings described in inflammatory breast tumors and provide the relationship between epigenetic changes and their contribution to the IBC origin and progression (Figure 1).

## 2. Clinical and Epidemiological Features

The clinical presentation of IBC combines the typical features of BC with inflammation signs. This tumor can be easily mistaken with other skin diseases and should be differentiated from noncancerous mastitis [5]. Delay in the accurate diagnosis and treatment can result in unfavorable clinical outcomes [19]. The inflammatory signals have a rapid onset, including erythema, edema, and enlargement of at least one-third of the breast. The skin over the breast is reddened, warmed, and thickened, with a peau d’orange appearance as a consequence of lymph vessels’ blockage by cancer cells in the skin. The diffuse feature of tumor emboli, formed by fewer adherent masses of cells, may contribute to high rates of metastatic disease at diagnosis and patients’ relapse [20,21]. The detection of tumor emboli invasion in dermal lymphatic vessels is an important feature, but its occurrence is not enough for the diagnosis. Dermal lymphatic invasion without typical clinical findings is not sufficient for a diagnosis of IBC [22].

The rarity of the disease and lack of consistency in databases and registries have hampered the establishment of IBC-associated risk factors [23]. However, racial differences have been reported as an independent predictor of elevated risk for an advanced stage at diagnosis, aggressiveness, and high mortality [24,25]. The analysis of incidence and survival trends for IBC based on the Surveillance, Epidemiology, and End Results (SEER) Program data (3648 cases) showed that African American women presented reduced survival compared to white women, both carrying IBC or LABC [4]. However, confounding factors such as socioeconomic status (health care access barriers), reproductive history, lifestyle, and diet could contribute to these differences reported in BC and IBC in African Americans compared to white American women [26]. Moreover, epidemiological studies have shown that IBC manifests at a younger age compared with non-IBC [4,21].

Socioeconomic status has also been associated as a risk factor for IBC development. The Florida Cancer Data System population-based study (1998–2002, based on 935 cases) reported longer median survival for patients living in more prosperous areas, using as criteria poverty level, insurance status, and race [27]. IBC women residing in areas with lower socioeconomic status present worse survival, which could be explained, at least in part, to differences in the treatment accessibility [28]. A recent study evaluated the relationship between IBC distribution and the impact of the social and economic area in the risk of developing the disease [29]. The authors found a geospatial clustering of IBC rates, with the larger cluster characterized by higher frequencies of unemployed individuals and poverty indexes. These findings suggested that additional environmental risk factors such as air, water, or soil contamination by pollutants may contribute to the IBC origin [29].

High body mass index (BMI) is an independent and positively associated risk factor for IBC. This finding is supported by the IBC Registry (based on 248 cases) at The University of Texas, MD Anderson Cancer Center, USA [30]. The overall mean BMI was 30.9 kg/m^2^, irrespective of menopausal status; 80.2% of patients showed BMI ≥ 25 kg/m^2^, and obesity in 82.6% of the African Americans [30]. In a large cohort (*N* = 617 cases), BMI was associated with increased risk of hormones receptor-positive and -negative in pre- and postmenopausal women, respectively [31]. Women with a positive history of breastfeeding showed a lower risk of IBC, while a higher proportion of IBC younger patients showed longer periods between menarche and the first birth [31]. Also, women at age <26 had a higher risk of triple-negative IBC [32,33].

Compared to non-IBC, IBC patients presented reduced disease-free survival (DFS) and overall survival (OS) [4]. The median OS for IBC patients is 4.75 years, while in non-IBC it is 13.40 years for stage III [29]. The DFS is 2.27 years for IBC versus 3.40 years for non-IBC stage IV patients [34]. Median survival was 2.9 years in IBC, shorter than that observed in women with LABC (6.4 years) or all other types of breast cancer (>10 years) [27]. A shorter survival was reported in African American women with IBC or LABC compared with white women with one of these tumors [4].

## 3. Molecular Profiling of Inflammatory Breast Carcinomas

Although IBC is considered a distinct clinicopathological entity, this disease shows significant heterogeneity at morphological (histological type, grade, and inflammatory infiltrate) and molecular (hormone receptor status and intrinsic molecular subtypes) levels [35]. The hormone receptor-positive subtype (estrogen and progesterone) is relatively rare in IBC (30%) compared to non-IBC (60–80%), and the subtypes HER-2 (erb-b2 receptor tyrosine kinase 2)-enriched and triple-negative are more frequent in IBC [36,37]. It is estimated that 40% of IBC are HER-2-positive tumors compared to 25% in non-IBC. The triple-negative subtype is detected in 30% of IBC versus 10–15% of non-IBC cases [4,36,37].

Genome-wide analysis of gene expression has indicated molecular features unique to IBC that may afford the prognostic stratification [35,38]. A 107-gene signature enriched for immunity-related genes was correlated with response to therapy [39]. However, the differences in the gene expression profiles are subtle and overlapped with non-IBC, failing to identify specific IBC molecular subtypes [40,41,42]. Independent of the molecular subtype found in IBC patients, the disease is more aggressive compared to non-IBC.

A complex pattern of aberrant DNA copy number alterations (CNA) has been reported in IBC in comparison with non-IBC samples. The genomic profile is characterized by a high level of genomic instability, frequently involving gains or amplifications [43,44]. At the mutational level, IBC harbored more mutations per sample than non-IBC cases [45]. These studies were based on target-next generation sequencing and detected somatic variants in well-known cancer driver genes, including *TP53*, *CDH1*, *NOTCH2*, *MYH9*, *BRCA2*, *ERBB4*, *POLE*, *FGFR3*, *ROS1*, *NOTCH4*, *LAMA2*, *EGFR*, *BRCA1*, *TP53, BP1*, *ESR1*, *THBS1*, *CASP8*, and *NOTCH1* [45,46]. Although efforts have been performed to characterize IBC at genomic levels, none of these studies resulted in a successful discovery of molecular signature or new therapeutic targets useful in the clinical practice.

## 4. Epigenetic of Mammary Gland Differentiation and Development

Epigenetics is a rapidly progressing field in biology in the last decades. Technological advances allowed the characterization of epigenetic patterns at high resolution, including the analysis of the functional effects of the epigenetic marks [47]. The “epigenetic revolution” benefited from the identification of protein families belonging to the epigenetic machinery. These proteins are grouped as writers or erasers since they can add or remove specific epigenetic marks in chromatin, respectively. The epigenetic marks are reversible and driven by enzymes showing opposite activities such as DNA methyltransferases (DNMTs) and demethylases (Ten-Eleven Translocation proteins, TET). Proteins involved in post-translational histone modifications, including histone lysine methyltransferases (HKMTs), histone lysine demethylases (HKDMs), histone lysine acetyltransferases (HATs), and histone deacetylases (HDACs) among others have been implicated in a range of biological processes. Permissive histone marks, including lysine acetylation and methylation (such as H3K9me1, H3K27me1, H3K36me3, H3K79me3, H3K4me1-3, and H4K20me1) are associated with chromatin accessibility, while lysine methylation is also a repressive mark and involved in the transcriptional silencing (for example, H3K9me3, H3K27me3, and H3K56me3) [48] (Figure 1).

The epigenetic alterations modulate the chromatin accessibility and regulate the gene expression during the differentiation of stem into differentiated cells [49], being essential for the transcriptional plasticity and tissue-specific cell differentiation throughout the mammary gland development [50,51]. The mammary gland shows several changes in its architecture throughout the lifetime, mainly in response to hormones. At birth, the gland is rudimentary, while at puberty, the estrogen causes duct-branching morphogenesis and ductal elongation. A secondary and tertiary side branching and the formation of alveolar units that produce and secrete milk occur during pregnancy and lactation. Finally, the mammary gland is reshaped to the quiescent state by the involution process (post-lactation weaning period) [51,52]. The deregulation of the epigenetic events involved in the mammary gland development and remodeling could result in the clonal proliferation of progenitor cancer cells in breast tissue driving breast carcinogenesis [53,54].

Recent studies have suggested that specific epigenetic profiles define commitment and multi-lineage differentiation of mammary stem cells. Accordingly, it has been hypothesized that bipotent mammary stem cells located in the basal compartment can give rise to both basal and luminal progenitor stem cells (CD44+) [51]. These progenitor cells can self-renew and, while basal progenitor stem cell differentiates in myoepithelial cells (CD10+), the luminal progenitor cells differentiate into alveolar or luminal lineages (CD24+ or MUC1+) [53]. Thus, changes in the methylome profile have the potential to contribute to the mammary cell identity and differentiation [55,56]. Genome-wide DNA methylation profiling analysis revealed that stem/progenitor cells are hypomethylated in comparison to luminal and myoepithelial cell types [57,58]. This finding supports the hypothesis that both expression of DNMTs and increase in DNA methylation are critical events during cell differentiation [59]. For instance, *HOXA1* and *TCF7L*, which are involved in stem cell maintenance, are hypomethylated and highly expressed [60], while epigenetic silencing (mediated by promoter methylation in stem/progenitor cells) and gene body methylation (associated with expressed genes as the luminal-driving transcription factor *GATA3*) are preferentially found in luminal cells [60,61].

Histone modifications also occur during mammary gland differentiation and development. During pregnancy, the luminal cells differentiate into alveolar cells (milk-secreting). At this stage, it is observed a decrease of repressive histone marks and an increase of the active marks in key luminal differentiation and milk-production genes [57,62]. In a mouse model, luminal progenitor cells show higher levels of the active H3K4Me3 (tri-methylation of lysine 4 of the histone H3) and lower levels of the repressive mark H3K27Me3 (tri-methylation of lysine 27 of histone H3) [63].

The establishment and maintenance of repressive histone marks, mediated by lysine methylases (KMTs) and demethylases (DKMTs), are crucial to cell identity. The repressive chromatin states are mediated by components of polycomb repressive complexes 1 and 2 (PRC1 and PRC2, respectively). For instance, the activity of EZH2, which is the catalytic subunit of PRC2, and the cognate levels of H3K27Me3 are influenced by progesterone during pregnancy and lead to multi-layered ducts and luminal cell hyperplasia [64]. In EZH2 (enhancer of zeste 2 polycomb repressive complex 2 subunit) knock-out mice models the depletion of luminal progenitor cells and delay in ducts’ formation occur [65]. Moreover, histone demethylation of cell-specific genes may also define cellular identity. The histone demethylase JARID1B, an enzyme that removes tri- and di-methylation of lysine 4 from H3 histone (H3K4), has been implicated in the mammary gland development by promoting luminal lineage-specific gene expression and repressing basal-specific genes [66].

## 5. Epigenetic Alterations in Inflammatory and Non-Inflammatory Breast Cancer

Progressive gains of promoter methylation resulting in silencing of tumor suppressor genes and global hypomethylation associated with genome instability have been extensively reported in BC [67,68]. Locus-specific aberrant DNA methylation patterns were correlated with several clinical, molecular, and pathological features, including the status of the protein p53, histological stage, and prognosis [69,70]. Hypermethylation of the *BRCA1* gene was associated with ER (estrogen receptor)-negative breast tumors and poor clinical outcome [71]. The expression levels of DNMT family members (DNMT3A, DNMT3B, and DNMT1) were associated with poor prognosis and low survival rates, supporting the previously reported oncogenic role [72].

Preclinical studies suggested that both gene mutation and epigenetic inactivation of *BRCA1* by promoter hypermethylation can cause protein down-regulation and subsequent sensitivity to PARP (Poly-ADP-ribose polymerase) inhibition [73,74]. Also, it is recognized that aberrant DNA methylation and changes in gene expression are correlated with the molecular subtypes of BC [9,75]. However, epigenetic studies in IBC are scarce. Van den Eynden et al. [76] were the first to report the overexpression of genes encoding caveolin-1 and -2 associated with hypomethylation of their promoter regions in the cell line SUM149, derived from IBC, in comparison to the human mammary epithelial cells (HMECs). A significantly increased expression of caveolin-1 and -2, both at the transcriptional and protein level, was also found in IBC compared to non-IBC [76]. Table 1 summarizes the studies describing DNA methylation in IBC samples.

DNA methylation microarray-based screening (27K Illumina Infinium Beadchip) was investigated to compare 19 IBC with 43 non-IBC samples [79]. The authors identified only four differentially methylated genes (*TJP3*, *MOGAT2*, *NTSR2*, and *AGT*), suggesting that abnormal DNA methylation is not the main force driving the molecular features of IBC [70]. A small number of studies have used methylation-specific polymerase chain reaction (MS-PCR)-based approaches and candidate genes [77,78,80]. Aberrant methylation in the promoter region of the *APC* gene was detected in 74% of IBC (*N* = 19) versus 46% of non-IBC (*N* = 35); nevertheless, *APC* mRNA and protein levels showed no significant differences according to the *APC* methylation status [77]. Subsequently, the same research group reported that *RARB* and *APC* genes were frequently hypermethylated in IBC (*N* = 19) compared to non-IBC (*N* = 81) [78]. Increased *GPX3* promoter hypermethylation was described in IBC (*N* = 20) compared to non-IBC (*N* = 20) samples [80]. The authors also correlated this aberrant epigenetic mark with *GPX3* down-expression at mRNA and protein levels.

Despite our limited knowledge of the epigenetic landscape of IBC, a compelling body of evidence has been accumulated, indicating that acquired epigenetic alterations are hallmarks of all stages of this disease. In addition to changes in DNA methylation patterns, IBC shows altered gene expression levels of epigenetics effectors such as histone modifiers. For instance, increased EZH2 protein expression was associated with lower local recurrence-free survival after radiotherapy in both BC and IBC [81]. Interestingly, EZH2 expression was inversely correlated with miRNA-26b expression in LABC and IBC [82]. These data suggested that EZH2 could be useful as a therapeutic target for these aggressive subtypes of BC.

Besides the DNA methylation and histone modification, aberrant expression of noncoding RNAs (microRNAs and long noncoding RNAs) has contributed to gene expression regulation in cancer [83]. Long noncoding RNAs (lncRNAs) interact with RNA-binding proteins in chromatin remodeling complexes and have emerged as potential key regulators of gene expression [84]. The lncRNA *DANCR* (differentiation antagonizing nonprotein coding RNA) was recently implicated in cancer-associated processes such as inflammation, inflammation-mediated epithelial-to-mesenchymal transition (EMT), and cancer stemness in late-stage triple-negative breast cancer [85]. The authors also demonstrated that EZH2 mediated the epigenetic mechanism leading to *SOCS3* down-regulation by the lncRNA *DANCR.* Based on these results, it was suggested that strategies suppressing *DANCR* or up-regulating *SOCS3* could prevent tumorigenesis and/or suppress the BC progression.

## 6. Evidence of Disrupted Epigenetic Program in the Inflammatory Breast Cancer Development

From the broadest standpoint view, IBC is a multifactorial disease in which genetic factors and environmental exposures interact to trigger its onset. Epigenetic mechanisms may mediate the complex epidemiologic relationships between genetic factors and the environment. At a mechanistic level, epigenetics could influence human diseases by three modes: (1) Epigenetics may be causally involved in a disease pathway by mediating genetic or environmental risk, (2) or be a risk modifier (genetic or environmental risk), or (3) by providing a mechanistic explanation to the genetic and environmental interactions in disease origin [86]. The scarcity of the disease hampers the development of epidemiologic studies, which is a limitation for the identification of epigenetic IBC-associated changes and to identify how these changes interact with risk factors, such as racial disparities and environmental influences. Herein, we described evidence linking epigenetic dysregulation and risk factors associated with IBC development as well as its role as a major player driving stemness phenotype, immunological escape, and metastatic potential during the tumor progression (Figure 2).

### 6.1. Racial Disparities and Epigenetic Alterations

Recent findings provide strong evidence for the epigenetic basis of racial health disparities in cancer [87]. Differences in normal breast tissue methylation patterns between African Americans and white American women have been reported, mainly involving cancer-associated genes [88]. Population-based cancer studies have demonstrated that race and ethnicity are associated with hereditary, dietary, environmental, lifestyle, and socioeconomic factors contributing to the epidemiological features of BC. Epigenetic patterns may also vary according to racial-ethnic identity and racial admixture [89,90]. Race and ethnicity reflect the shared environmental exposures, which, in turn, also have a significant impact on the epigenetic profiles and risk for human diseases. However, it has been suggested that stratification by genetic ancestry could improve the precision of population-based studies. Galanter et al. [91] described that both self-identified ethnicity and genetically determined ancestry were significantly associated with the DNA methylation profile in blood samples of 573 individuals of Hispanic origin. The authors suggested that differential methylation between ethnic groups could be partially explained by the shared genetic ancestry. However, the enrichment of DNA methylation in sites ethnicity-associated among loci previously related to environmental exposures, not captured by ancestry, would also explain the racial/ethnic disparities in the incidence of human diseases [91].

Recently, Fouad and colleagues [30] identified a distinct epidemiological profile between African Americans and white women with IBC. These patients differed in the risk of developing IBC according to reproductive history, breastfeeding, and obesity. African American young patients giving birth to their first child and showing multiparity, presented a significantly high risk to develop IBC. The authors suggested that a shorter period between age at menarche and of giving their first birth and lack of breastfeeding create a cancer-prone microenvironment in the breast. During pregnancy, a breast remodeling, an increase in mammary progenitor cells and pro- and anti-inflammatory mediators occur, which should drop after weaning. However, lack of breastfeeding would lead to the accumulation of a pro-inflammatory microenvironment and the production of oxidative stress, leading to DNA damage. For white patients, the reported risk factors were older age at first birth, a fewer number of pregnancies associated with a longer period between age at menarche and age at first birth. These risk factors were associated with breast tissue aging and the accumulation of molecular damage. The authors also suggested that during pregnancy, the breast remodeling, hormonal induction of cell proliferation, and the accumulated molecular damage result in clonal expansion of altered cells and tumorigenesis [32].

### 6.2. Obesity and Body Mass Index (BMI)

Overweight and obesity increase the risk of several cancer types, including BC. This phenotype is also associated with worse prognosis after cancer diagnosis and treatment [92]. Body mass index (BMI) is recognized as an independent risk factor for IBC development [34]. However, it is under debate if BMI itself or obesity-associated health behaviors, like diet and exercise, play a role in breast cancer etiology [93]. The relationship among obesity, diet, and physical activity are poorly understood. One of the promising pathways that underlie this association is inflammation, a known hallmark of cancer [94]. Furthermore, diet and physical activity may reduce systemic inflammation [95,96].

A growing number of studies has reported the modifying effects of diet and obesity on DNA methylation patterns of genes associated with BC risk and patients’ outcome [97]. Differential variability in genome-wide methylation profiles was observed as a feature of both BC and obesity [98]. Phase I of Carolina Breast Cancer Study (*N* = 345 BC) reported that DNA methylation levels at CpG dinucleotides tend to increase with obesity (BMI ≥ 30). This study indicated that obesity was associated with DNA methylation changes in genes involved in the immune response, cell growth, and DNA repair [99]. The clinical trial identifier NCT00811824 evaluated the impact of lifestyle changes (diet and weight loss) on global CpG methylation in peripheral blood from Hispanic, African American, and Afro-Caribbean women on BC survivors (clinicaltrials.gov). This study demonstrated that changes in dietary habits and other lifestyle factors were associated with DNA methylation changes [100]. In summary, DNA methylation biomarkers should be useful to monitor lifestyle interventions, and differentially methylated sites obesity-associated could serve as targets for BC prevention [101].

### 6.3. Environmental Exposure to Endocrine Disruptors

As mentioned above, IBC shows an asymmetric geospatial distribution, tending to cluster in regions with high rates of unemployment and poverty, which reinforces the association between this disease and environmental factors [29]. Environmental factors can modulate cytokines, growth factors, hormone levels, and the release of molecules associated with stress-response and neurotropic factors [102]. Indeed, the environment influences the epigenome, and the epigenetic and genetic background of each individual can influence specific responses to these environmental factors. Exercise, microbiome, alternative medicine, exposure to toxic chemicals and drug addiction, social interactions, psychological state, diet, seasonal changes, financial status, therapeutic drugs, and positive history of diseases might have beneficial or harmful effects depending on the specific nature of those influences [102].

The accumulation of environmental compounds with influence in endocrine signaling pathways due to its mimetic action of endogenous molecules is a potential factor involved in the epigenetic modifications of IBC. These molecules, called “endocrine disrupting chemicals” (EDCs) [103], could affect the offspring through epigenetic mechanisms [104], including mammary glands in female offspring [105]. Bisphenol A (BPA), a compound used in the manufacture of polycarbonate and epoxy resin, is one of the most common EDCs found in the environment, detected in water, soil, and air. Humans may consume BPA through canned food since epoxy resin that coats food cans releases BPA [103]. BPA-based products can reach the rivers, marine waters, the effluent treatment, and landfill sites [106].

Several studies describe the effect of BPA on epigenetic programming during embryonic development, including examples of altered gene expression modulated by histone modifications, DNA methylation changes, and noncoding RNA. In developmental in vivo models, BPA exposure increases the susceptibility to prostate cancer [29] and is potentially associated with breast carcinogenesis [107]. In experimental mice models, the continuous exposure to BPA before mating and during gestation and lactation resulted in DNA hypomethylation, obesity, diabetes, and increased tumorigenicity in the offspring [108]. Indeed, law regulations set to decrease BPA have been established worldwide, resulting in decreased levels in the child population since 2011. However, the corresponding intake in the adult population continued to increase. This difference may be the result of the prohibition of the use of BPA in food-related products for children in many countries since 2009 [109].

Synthetic hormones, insecticides, and pesticides widely utilized can also accumulate in the environment and behave as EDCs. For example, persistent epigenetic changes induced by diethylstilbestrol (DES) exposure can be passed to the next generation [110]. DES is a synthetic estrogen antagonist that in animal models induces vaginal cancer in the offspring of the treated mothers [102]. Interestingly, in utero exposure to DES increases EZH2 expression in the mammary gland, providing a mechanistic explanation linking endocrine disruptors and BC [107].

Insecticides and pesticides, like dichlorodiphenyltrichloroethane (DTT), methoxychlor, vinclozolin, and permethrin, have several effects on the epigenome. These chemicals can alter the DNA methylation and the level of expression of the DNA methyltransferase DNMT3B [111]. The exposure to methoxychlor influences the expression levels of several genes, including the paternally imprinted *H19*, *GTL2*, and *MEG3* and the maternally imprinted genes *PEG1*, *PEG3*, and *SNRP*. These effects are mediated by altered CpG methylation levels, with the effect persisting through three generations in rats [112].

Analysis of chemical compounds in enriched IBC world regions, where the rates of the disease are high, could be an interesting strategy to establish prevention protocols. For example, as continuous exposition to BPA leads to DNA hypomethylation in mice, it has been suggested that its adverse effects could be compensated by dietary supplementation with methyl donors [108].

### 6.4. Stemness and Microenvironment

Stem cell-like tumor phenotype (stemness), epithelial-to-mesenchymal transition (EMT), and specific transcriptional patterns are reported in IBC cells [6]. As described above, substantial evidence suggests that epigenetic mechanisms are orchestrated to maintain undifferentiated cells or to establish multi-lineage commitment/differentiation states in the mammary tissue [51,52,53]. Furthermore, it has been proposed that the cellular differentiation is not unidirectional upon stress, injury, or experimental stimuli and that ‘terminally differentiated’ cells may exhibit plasticity [113,114].

Mammary stem or progenitor cells could be a target for cellular transformation, once they have a long-span life and are susceptible to molecular injuries due to persistent exposure to several environmental agents and inflammation [115,116]. The xenoestrogens may mimic estrogenic actions that aberrantly influence epithelial differentiation in mammary glands [117,118]. Also, epithelial cells derived from estrogen-exposed breast progenitor cells show methylome alterations similar to BC, suggesting an association between estrogen injury of breast stem/progenitor cells and carcinogenesis [117]. Xenoestrogens may also stimulate the phosphorylation of membrane-bound proteins activating different kinase signal transduction pathways required for transcriptional regulation [119]. Chronic exposure to endocrine disruptors has been related to transcriptional reprogramming of breast cancer cell lines by altering the physiological pathways of hormone signaling permanently. This exposition promotes changes in the expression levels of estrogen-responsive target genes, estrogen receptor (ER) recruitment, the levels of H3K4me, and an increase of deoxyribonuclease accessibility at the cis-regulatory sites of response elements [120]. Both estrogen and glucocorticoid receptors have been implicated in breast cancer development. The crosstalk between these receptors modulates the access of ER to specific sites in the genome by the reorganization of the chromatin conformation [121]. Besides, the trans-activation by steroid hormones involves chromatin remodeling mediated by glucocorticoid receptors, such as BRG1, before the recruitment of FOXA1 and GATA3 [122]. GATA3 is a key transcription factor of mammary epithelial cell differentiation and was implicated in the chromatin reprogramming during the mesenchymal-to-epithelial transition. In this process, GATA3 binds to chromatin and recruits co-factors (such as BRG1) and the SWI/SNF (SWItch/Sucrose Non-Fermentable) chromatin remodeling complex, which results in nucleosome eviction and histone modifications [123]. The network of interactions between steroid hormones receptors and transcription factors modulates the chromatin landscape and determines a plethora of transcriptional responses. These interactions are complex and not characterized in-depth, but are crucial to advance our knowledge of chromatin dynamics during breast cancer progression [124].

Epigenetic modifications may regulate growth and differentiation of pluripotent stem cells by DNA methylation, depletion of histone H1, and changes of histone marks at the promoter region of the gene encoding the octamer-binding transcription factor (*OCT*). All these changes drive the cells to commit to a particular lineage by repressing genes associated with differentiation to alternative lineages. In this context, alteration in the epigenetic balance (the combination of histone modifications and DNA methylation) could provide the necessary switch for the formation of cancer stem cells [125].

Aberrant activation of the Notch signaling pathway has been described in several human cancer types. The highly conserved family of Notch receptors induces cell proliferation, metastasis, and epithelial-mesenchymal transition [126]. The activation of Notch 3 was implicated in the IBC development based on the human xenograft MARY-X model in vivo and corresponding spheroids in vitro [127]. These studies showed that selective Notch 3 activation was correlated with increased expression of the intracellular domain of Notch 3 (N3icd) and six target genes (*HES-5*, *HEY-1*, *c-MYC*, *DELTEX-1*, *NRARP*, and *PBX1*). Although in vivo inhibition of Notch 3 with gamma-secretase inhibitors (GSIs) resulted in the diffusion-limited effects on Notch 3 signaling, a reduction of xenograft growth was observed [127]. This finding indicated that Notch 3 might be a therapeutic target for IBC. Increased HKDMs’ expression was associated with unfavorable prognosis, activation of genetic programs of cell proliferation, luminal to basal-like transition, progression, and metastasis [128]. KDM2A regulates Notch signaling, stemness, and angiogenesis in BC [128]. Although KDM2A expression is unknown in IBC, the expression of genes related to the Notch pathway should be considered in the treatment strategies targeting Notch signal pathways.

### 6.5. Inflammatory Pathways

Inflammatory and immune response IBC-associated pathways have provided new insights into the role of innate or adaptive immune cells in the regulation of tumor progression. Preclinical and clinical studies have suggested that immunotherapy has the potential to improve the clinical outcome [129]. Among the immune response pathways, the TGFβ pathway is often attenuated in IBC samples [40]. Overexpression of TGFβ may disrupt clustering and invasion of IBC cells, suggesting that decreased TGFβ levels are essential in tumor emboli formation [130]. TGFβ1 and its receptor are dysregulated by promoter methylation in several tumor types [131,132].

Interleukin-6 (IL-6), one of the major cytokines in the tumor microenvironment, is involved in the host’s immune defense system and in the modulation of growth through an autocrine and paracrine secretion and differentiation [133]. IL-6 overexpression has been associated with tumor progression, inhibition of cancer cell apoptosis, stimulation of angiogenesis, and drug resistance. In IBC, the IL-6 pathway is frequently hyperactivated; nonetheless, if suppressed, tumor cells repress E-cadherin (epithelial) expression [134]. IL-6 was associated with IBC development and induced migration of MSCs (mesenchymal stem cell) [134]. Epigenetic modification of IL-6 has been reported in several cancer types [135,136,137]. In breast cancer, p53 deficiency induces an epigenetic reprogramming IL-6-dependent by methylation changes, driving tumor cells to basal-like and stemness phenotype [138].

The nuclear factor-κB (NFKB) transcription factor family is activated by various intra- and extra-cellular stimuli such as cytokines, oxidant-free radicals, among others. Activated NFKB regulates the expression of genes directly involved in the synthesis of inflammatory cytokines, chemokines, and adhesion molecules [139]. In the canonical NFKB pathway, expression of genes involved in cell proliferation, survival, innate immunity, inflammation, and angiogenesis are modulated, whereas the noncanonical pathway regulates the homeostasis of adaptive immunity and lymphangiogenesis. The NFKB pathway acts as an inflammatory mediator involved in EMT (epithelial-mesenchymal transition) in IBC tumors, associated with resistance to chemotherapy and endocrine therapy via evasion of apoptosis [140]. NFKB can be methylated on both lysine and arginine residues by histone-modifying enzymes, including lysine and arginine methyltransferases and demethylases [141]. These epigenetic modifications found in NFKB is an interesting target for new therapeutic approaches with epigenetic drugs.

## 7. Epigenetic Therapy: The Inhibitors of Histone Deacetylases (HDACi)

Unlike genetic changes, epigenetic alterations are potentially reversible, being more prone to corrective therapeutic reversal. Epigenetic therapies are becoming a promising alternative to improve the average response rates to cancer treatment acting in synergy with other therapeutic modalities or to reverse acquired resistance to chemo-radiotherapy [142]. A few epigenetic drugs were approved by the U.S. Food and Drug Administration (FDA). The so-called epi-drugs include DNMT inhibitors (DNMTi), such as cytidine analogs, which can induce DNA demethylation, i.e., azacitidine (Vidaza) and 5-aza-2′-deoxycytidine (decitabine). The vorinostat, panobinostat, belinostat, and romidepsin are FDA-approved HDAC inhibitors (HDACi). These last drugs act by blocking the catalytic domain of HADCs and are associated with changes in the acetylation patterns of histones [143]. Although the undesirable off-target effects of epi-drugs still need to be overcome, accumulating findings pointed out that this therapy may reverse stem cell-like behavior and chemoresistance [144]. The treatment with epi-drugs affects multiple cancer-associated processes, including cell signaling, survival, apoptosis, DNA damage repair, and immune response and evasion [145,146]. The role of epigenetic mechanism in modulating immune cell function and antitumor immunity also supports that epigenetic therapy may be combined with immunotherapy in cancer at advanced stages [147].

Changes in histones acetylation and deacetylation, mediated by a family of histone acetyltransferases (HATs) and deacetylases (HDACs), respectively, have been described in human cancers, including BC [47]. HDACs belong to histone deacetylase family (arginase/deacetylase superfamily) or the Sir2 regulator family (deoxyhypusine synthase-like NAD/FAD-binding domain superfamily). These enzymes remove the acetyl group from lysine residues and are grouped into four classes (I–IV). HDACs classes I, II, and IV need a zinc ion (Zn^2+^) and share a similar catalytic core for acetyl-lysine hydrolysis, while class III requires a nicotinamide adenine dinucleotide for its enzyme activity [148]. HDACs may act on nonhistone subtracts by deacetylating other proteins, including transcription factors and integrators of cell signaling that are cellular signal transducers, like cytokines and growth factors (TGFβ, EGF, interferon, and WNT signaling). Many nonhistone proteins affected by HDACs are oncogenic and drivers of the cancer onset, incidence, progression, and metastasis, such as E2F, p53, c-MYC, NFKB, SMAD7, PTEN, β-catenin, STAT, GATA, and FOXO [72].

The effects of the treatment with the HDAC inhibitors, CG-1521 and trichostatin A, were investigated in two cell lines (SUM149PT and SUM190PT) [149]. Only 9% of the genes were modulated in both cell lines after global gene expression analysis. SUM149PT cells (*TP53* mutated, *BRCA1* wild type) are derived from a triple-negative breast cancer cell line (derived from a mouse xenograft of a transplanted primary human invasive infiltrating ductal carcinoma metastatic nodule). SUM190PT cell line is derived from ER- and PR (progesterone)-negative primary IBC and characterized for having *TP53* p.Gln317Ter (c.949C>T) variant, HER2 overexpression, and *BRCA1* wild type [150]. The authors suggested that these HDACi target different biological processes in these cell lines. Alternatively, the epi-drugs may inhibit different HDACs enzymes in these cell lines [149].

RNAi-based loss-of-function studies at a genome-wide level revealed that HDAC6 is necessary to maintain IBC cell viability [151]. The authors reported that ricolinostat (HDAC6 inhibitor) controls IBC cell proliferation both in vitro and in vivo. Treatment with the HDACi entinostat in combination with HER2-targeted agents increased apoptosis in trastuzumab-resistant IBC cells [152]. The HDACi vorinostat and valproic acid effectively induced apoptosis of cancer cells and may be used in the TNBC treatment, including TN-IBC. This apoptotic activity was dependent on the pentose phosphate pathway and may be targeted in IBC with combined HDAC and glucose-6-phosphate 1-dehydrogenase (G6PD) inhibition [153]. A recent novel combination treatment of histone deacetylase (HDAC) type I and MAP2K7 MEK (mitogen-activated protein kinase kinase 7) inhibitors, entinostat and pimasertib, was used to treat IBC and TNBC xenograft models both in vitro and in vivo. These studies revealed significant synergistic antitumor activity of entinostat and pimasertib by effective degradation of MCL1 (MCL1 apoptosis regulator, BCL2 family member) through increased NOXA expression and apoptosis [154]. NOXA (NADPH oxidase activator 1) /PMAIP1 (phorbol-12-myristate-13- acetate-induced protein 1) is a member of the BCL-2 (BCL2 apoptosis regulator) family detected in 65% of IBC and TNBC cell lines after entinostat treatment. NOXA/PMAIP1 promotes the degradation of MCL1, a protein member of the BCL-2 family, with an anti-apoptotic role [154].

Recently, a phase II clinical trial in metastatic IBC (NCT01938833) is using Romidepsin (a selective inhibitor of histone deacetylase) and Abraxane (an albumin-bound form of paclitaxel). Phase I/II clinical trials for the treatment of acute leukemia and non-Hodgkin lymphoma are ongoing with HKMT inhibitors [155]. EZH2 inhibitors have been tested in relapsed or refractory B-cell non-Hodgkin’s lymphoma and malignant rhabdoid tumors. As mentioned above, EZH2 is overexpressed in IBC and correlated with unfavorable prognosis [81]. Therefore, this new drug is a candidate to be tested in inflammatory breast tumors.

## 8. Final Remarks and Conclusions

Currently, the main challenges in the oncological practice to the management of IBC patients are the prompt recognition of the disease symptoms at diagnosis and the identification of useful markers to guide treatment decisions and improve the rates of therapy response. This intriguing type of breast cancer is rare and lethal, and the number of studies devoted to profiling its biological features at mutational, transcriptomic, and epigenomic levels are scarce. The molecular characterization is essential to advance our understanding of IBC biology and translate the knowledge to refine the diagnosis and to improve therapeutic management.

Epigenetic alterations, such as DNA hypermethylation and changes of histone modifications patterns, are a common denominator among all human cancers. Aberrant epigenetic marks are labile, dynamic, and selected throughout the carcinogenic process. These alterations are identified at the early stage of pre-malignant conditions and during cancer progression. In this review, we summarized the current molecular knowledge that links IBC origin and progression to a disrupted epigenetic program. Epigenetic changes are associated with well-established risk factors for IBC and should be a link to identify at-risk patient populations and to test chemopreventive strategies. Importantly, epigenetic alterations mediate the functional properties of tumor cells, including self-renewing capacity, recurrence, metastatic potential, and resistance to chemotherapy [156]. The reversible nature of epigenetic modifications provides a unique opportunity for the development of epigenetic therapy. A compelling body of evidence supports that proteins of epigenetic machinery and their marks are promising targets for the development of new therapeutic approaches for IBC patients. Epi-drugs can modulate the sensitivity of cancer cells to treatment and should be considered in multimodality chemotherapy and immunotherapy protocols. In addition to proteins that write or erase epigenetic marks, the class of protein termed readers has emerged as new potential targets for epigenetic therapy [157]. Bromodomain and extra terminal domain (BET) proteins are implicated as key factors leading to increased expression of genes involved in carcinogenesis. Thus, BET inhibitors have been included in several ongoing clinical trials of hematological and solid tumors [158], including breast cancer [159].

Efforts in the global characterization of the epigenetic landscape of IBC must be implemented to accelerate the translation of this knowledge into clinical practice. Given its rarity and limited resources, multicenter collaborative research efforts are needed.

## Figures and Tables

**Figure 1 cells-09-01164-f001:**
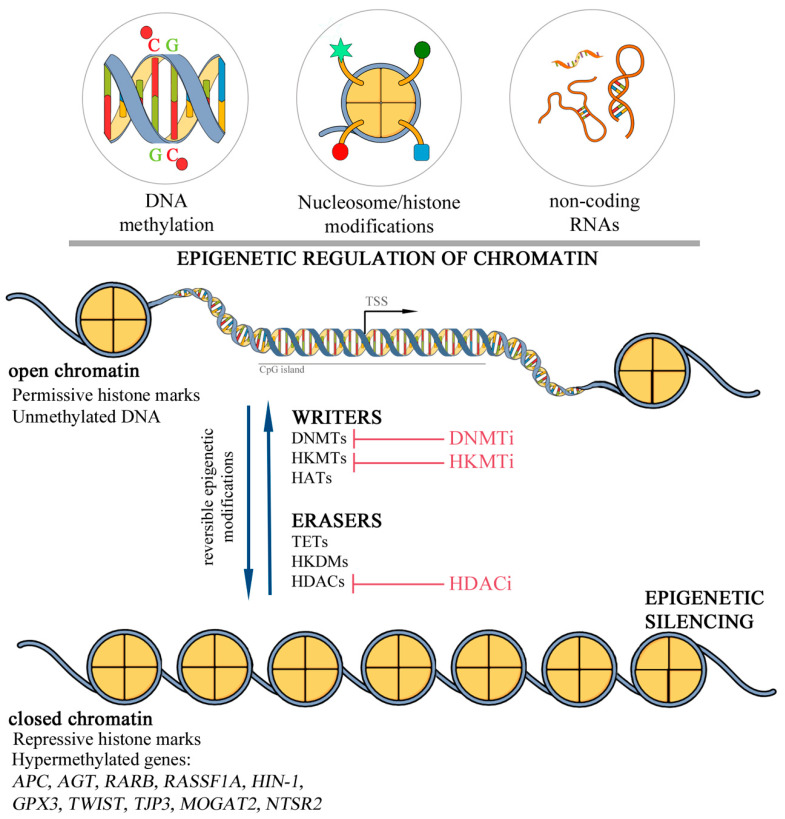
Epigenetic regulation of gene expression. (**A**) DNA methylation, histone modifications, and noncoding RNAs modulate the chromatin accessibility and gene expression. In cancer, dysregulated histone marks and DNA hypermethylation contribute to epigenetic silencing of tumor suppressor genes. Epigenetic regulation is a dynamic and reversible process mediated by enzymes termed “writers” and “erasers”. Inherent reversibility of epigenetic changes and aberrant epigenetic marks are promising anticancer strategies based on inhibitors targeting these enzymes (DNMTs—DNA methyltransferases, HATs—histone acetyltransferases, HKMTs—histone lysine methyltransferases, HDACs—histone deacetylases, HKDMs—histone lysine demethylases, TETs—ten-eleven translocation family proteins, TSS—transcription start site, DNMTi, HKMTi, HDACi—inhibitors of DNMTs, HKMTs or HDACs, respectively). The image was performed with Mind the Graph (mindthegraph.com).

**Figure 2 cells-09-01164-f002:**
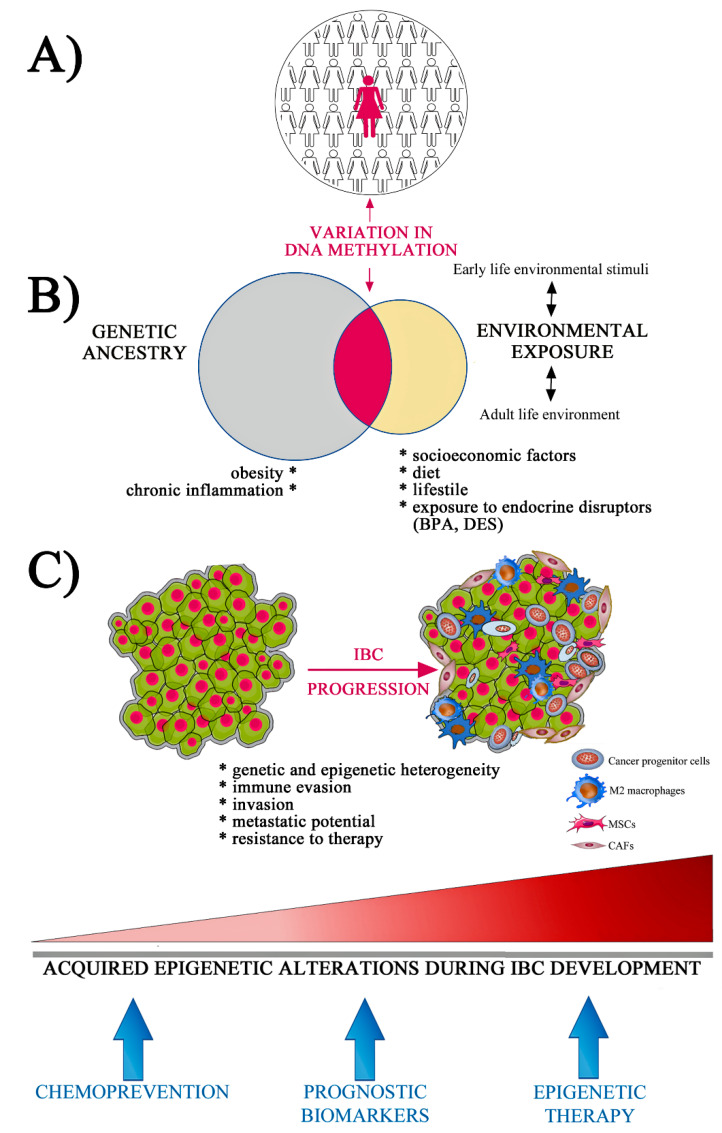
(**A**) Effects of genetic ancestry and environmental exposures on racial/ethnic disparities of IBC (inflammatory breast cancer) incidence. (**B**) Changes in DNA methylation have been associated with risk factors such as obesity, chronic inflammation, and environmental exposure. Venn diagram shows the overlapping between genomic ancestry and environmental exposure on differential methylation in ethnic subgroups associated with the risk of IBC development (central area in dark red). (**C**) Accumulation of genetic and epigenetic alterations drive tumor progression and contribute to the heterogeneity and plasticity of cancer cells. Epigenetic changes also provide new opportunities for chemoprevention and biomarkers’ development. (CAFs, cancer-associated fibroblasts; MSCs, mesenchymal stem cells; BPA, bisphenol A; DES, diethylstilbestrol). The image was performed using Mind the Graph (mindthegraph.com).

**Table 1 cells-09-01164-t001:** Summary of DNA methylation alterations in inflammatory breast cancer.

Target Gene(s)	Strategy	Sample Size	Main Findings	Refs.
*APC*	Quantitative Methylation-Specific PCR	21 IBC/ 35 Non-IBC	High methylation levels in IBC versus non-IBC	[77,78]
*APC* *DAPK* *HIN-1 RASSF1A RARB* *TWIST1*	Quantitative Methylation-Specific PCR using a six-gene panel	19 IBC/ 81 Non-IBC	Differential methylation of *RARB* in IBC *RARB* and *APC* methylation frequency were significantly increased in IBC	[78]
Array-based DNA methylation profiling	Illumina Infinium Human Methylation 27 Beadchip	19 IBC/ 43 non-IBC	Four-gene based signature (*TJP3*, *MOGAT2*, *NTSR2*, and *AGT*) differentiates IBC from non-IBC	[79]
*GPX3*	qualitative Methylation-Specific PCR	20 IBC/ 20 non-IBC	High promoter hypermethylation of *GPX3* gene in IBC versus non-IBC	[80]

*APC*, APC regulator of WNT signaling pathway; *DAPK*, death associated protein kinase 1, *HIN-1*, secretoglobin family 3A member 1; *RASSF1A*, Ras association domain family member 1, transcript A; *RARB*, retinoic acid receptor beta; *TWIST1*, twist family bHLH transcription factor 1; *TJP3*, tight junction protein 3; *MOGAT2*, monoacylglycerol *O*-acyltransferase 2; *NTSR2*, neurotensin receptor 2; *AGT*, angiotensinogen; *GPX3*, glutathione peroxidase 3.

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
