# Peer review of "Epigenetics in Inflammatory Breast Cancer: Biological Features and Therapeutic Perspectives"

_cells, 2020, doi:10.3390/cells9051164_

Round 1
Reviewer 1 Report
This is a comprehensive review highlighting the role of epigenetic modulators in inflammatory breast cancer and the potential for targeting epigenetic mechanisms for treatment. This is an important area of research and the review is generally well-prepared. Overall, this review is well-organized and covers many aspects related to IBC. While comprehensive, this causes it to stray from the focus. While the title and the abstract suggest that the focus is on epigenetics in IBC, the review almost covers too much and the focus is lost. In particular, while the sub-topics under section 7 of the review all discuss important topics in understanding the prevelance and progression of IBC, the links to epigenetics are quite cursory in some instances. These sections could either be shortened or the focus could be more directly linked to the epigenetic observations (specifically DNA methylation and histone modifications) associated with each of these factors (racial disparities, environmental influences, etc). While there may not be much data on some of these aspects, this could be stated briefly so as not to divert the focus of the review. The only other recommended change is to make Figure 1B, in particular, more specific. Can the authors add to their figure some of the epigenetic changes that occur during IBC development? This would help to highlight some of the observations about epigenetics in IBC that are discussed in the text.
Author Response
Point-by-point response to Reviewer 1 comments
Comments and Suggestions for Authors
This is a comprehensive review highlighting the role of epigenetic modulators in inflammatory breast cancer and the potential for targeting epigenetic mechanisms for treatment. This is an important area of research and the review is generally well-prepared. Overall, this review is well-organized and covers many aspects related to IBC.
Comment 1. While comprehensive, this causes it to stray from the focus. While the title and the abstract suggest that the focus is on epigenetics in IBC, the review almost covers too much and the focus is lost. In particular, while the sub-topics under section 7 of the review all discuss important topics in understanding the prevalence and progression of IBC, the links to epigenetics are quite cursory in some instances. These sections could either be shortened or the focus could be more directly linked to the epigenetic observations (specifically DNA methylation and histone modifications) associated with each of these factors (racial disparities, environmental influences, etc). While there may not be much data on some of these aspects, this could be stated briefly so as not to divert the focus of the review.
Response: Given that inflammatory breast cancer (IBC) is a rare and poorly studied disease when compared to other breast cancer subtypes, we presented a comprehensive review covering two main points. First, we described how the epigenetic mechanisms are disrupted during IBC genesis and progression and how epigenetic changes could be modulated or associated with the risk factors already reported in the IBC development. Second, consistent evidence linking epigenetic changes with recognized risk factors IBC-associated is poorly explored in literature, which is a limiting factor in organizing the subtopics of section 7 (racial disparities, environmental influences, among others). As recommended, this limitation was stated in the first paragraph of section 7 (lines 267 - 273), which was edited according to the three reviewers’ suggestions.
Comment 2. The only other recommended change is to make Figure 1B, in particular, more specific. Can the authors add to their figure some of the epigenetic changes that occur during IBC development? This would help to highlight some of the observations about epigenetics in IBC that are discussed in the text.
Response: In the current version, we split the figure 1 in two figures (Figures 1 and 2) showing more details of the epigenetic alterations and the risk factors IBC-associated.

Reviewer 2 Report
This is a scientifically sound, well written paper. I have no major comments and highly recommend this paper be accepted.
Minor comments:
Line 52: "Epigenetic mechanisms mediated by" should be changed to "epigenetic mechanisms such as" Line 57: "silencing of cancer driver genes"-->The relationship is bidirectional it can both turn on and turn off genes improperly. Please change the text to reflect this Lines 78-80: SEER found that AA women had generally reduced survival relative to their white counterparts, suggesting race-based bias in care/the health care system. This context is also important to remember so touching on it briefly is important. Lines 275-278: Race is a social construct, so it should be addressed whether women were grouped by self-identified ancestry vs. genetic ancestry markers. Additionally, particularly in the US, race is linked to differences in environmental exposure which can alter epigenetic profiles, so it is unclear whether differences are due to ancestry alone vs. environmental exposures. This should be addressed in the racial disparities section. Lines 374-381: Other hormones could potentially play a role (testosterone, cortisol, progesterone) as well as estrogen. There is little literature but touching on some of the work from Trevor Archer's Lab and Gordon Hagar's lab would help expand this section.
Author Response
Point-by-point response to Reviewer 2 comments
Comments and Suggestions for Authors
This is a scientifically sound, well written paper. I have no major comments and highly recommend this paper be accepted.
Minor comments:
Comment 1:
Line 52: "Epigenetic mechanisms mediated by" should be changed to "epigenetic mechanisms such as"
Line 57: "silencing of cancer driver genes"-->The relationship is bidirectional it can both turn on and turn off genes improperly. Please change the text to reflect this.
Response: As recommended, we modified these lines in the current version of our manuscript.
Comment 2. Lines 78-80: SEER found that AA women had generally reduced survival relative to their white counterparts, suggesting race-based bias in care/the health care system. This context is also important to remember so touching on it briefly is important.
Response: Thank you for the suggestion. We have included the race-based bias in lines 81-84.
Comment 3. Lines 275-278: Race is a social construct, so it should be addressed whether women were grouped by self-identified ancestry vs. genetic ancestry markers. Additionally, particularly in the US, race is linked to differences in environmental exposure which can alter epigenetic profiles, so it is unclear whether differences are due to ancestry alone vs. environmental exposures. This should be addressed in the racial disparities section.
Response: This issue was addressed in lines 293-302, as suggested.
Comment 4. Lines 374-381: Other hormones could potentially play a role (testosterone, cortisol, progesterone) as well as estrogen. There is little literature but touching on some of the work from Trevor Archer's Lab and Gordon Hagar's lab would help expand this section.
Response: Thank you for the suggestion. We have revised the literature and included additional information in lines 403-419.

Reviewer 3 Report
This review article by Costa Faldoni and co-workers aims to revise available data on epigenetics involvement in inflammatory breast cancer, a very rare and particularly aggressive disease.
The review is very clear and well-written, comprehensively covering various aspects of inflammatory breast cancer, with special attention to epigenetics-linked mechanisms.
The only weak point that can be improved is the complete lack of discussion of epigenetic “readers” among the proteins capable of integrate epigenetic signals. These proteins (e.g. bromodomain, chromodomain…) are involved in many cancer types, including breast, and are also target of therapeutic compounds. For these reasons chromatin readers should be included and discussed.
Very few typos or unclear sentences, for example:
Lines 153-154: revise sentence, maybe verb form not correct
Line 279: for -> by
Lines 303-304: revise sentence, it seems that the meaning is: “inflammation causes inflammation”
Lines 486-487: revise sentence, maybe removing “A” at the beginning
Line 510: promisor->promising
Author Response
Point-by-point response to Reviewer 3 comments
Comments and Suggestions for Authors
This review article by Costa Faldoni and co-workers aims to revise available data on epigenetics involvement in inflammatory breast cancer, a very rare and particularly aggressive disease. The review is very clear and well-written, comprehensively covering various aspects of inflammatory breast cancer, with special attention to epigenetics-linked mechanisms.
Comment 1.
The only weak point that can be improved is the complete lack of discussion of epigenetic “readers” among the proteins capable of integrate epigenetic signals. These proteins (e.g. bromodomain, chromodomain…) are involved in many cancer types, including breast, and are also target of therapeutic compounds. For these reasons chromatin readers should be included and discussed.
Response: The bromodomain inhibitors are promising drugs to be used in the treatment of IBC. In a recent review article published by Khandekar & Tiriveedhi [Cancers, 2020 Mar 25;12(4)], it is stated that “preliminary evidence for clinical studies does not support the application of BET inhibitors as monotherapy in cancer treatment and the combinatorial of BET inhibitors with other chemo-and immunotherapeutic agents remain elusive in triple-negative breast cancer”. Based on this preliminary evidence, we include the potential use of BET inhibitors in the perspective section, lines 550-555.
Comment 2. Very few typos or unclear sentences, for example:
Lines 153-154: revise sentence, maybe verb form not correct
Line 279: for -> by
Lines 303-304: revise sentence, it seems that the meaning is: “inflammation causes inflammation”
Lines 486-487: revise sentence, maybe removing “A” at the beginning
Line 510: promisor->promising
Response: The suggestions were accepted, and the typos were corrected (highlighted in gray).